# Electrical stimulation of smiling muscles reduces visual processing load and enhances happiness perception in neutral faces
J. Baker [1,2] ✉, HVV Ngo [1], T. N. Efthimiou [1,3], A. Elsenaar [4], M. Mehu [5] & S. Korb [1,6] ✉

Theories of embodied cognition suggest that after an initial visual processing stage, emotional faces elicit spontaneous facial mimicry and that the accompanying change in proprioceptive facial feedback contributes to facial emotion recognition. However, this temporal sequence has not yet been properly tested, given the lack of methods allowing to manipulate or interfere with facial muscle activity at specific time points. The current study (N = 51, 28 women) investigated this key question using EEG and facial neuromuscular electrical stimulation (fNMES)—a technique offering superior control over which facial muscles are activated and when. Participants categorised neutral, happy and sad avatar faces as either happy or sad and received fNMES (except in the control condition) to bilateral zygomaticus major muscles during early visual processing (−250 to +250 ms of face onset), or later visual processing, when mimicry typically arises (500–1000 ms after face onset). Both early and late fNMES resulted in a happiness bias specific to neutral faces, which was mediated by a reduced N170 in the early window. In contrast, a modulation of the beta-band (13–22 Hz) coherence between somatomotor and occipital cortices was found in the late fNMES, although this did not predict categorisation choice. We propose that facial feedback biases emotion recognition at different visual processing stages by reducing visual processing load.

According to theories of embodied cognition[1], emotion recognition and affective judgments are supported by humans' ability to simulate, both in the brain and body, observed emotional states[2–4]. In line with this, emotional faces automatically trigger somatomotor activity in the brain[5,6], and spontaneous facial mimicry in the face[7], while interference with either can impair emotion recognition and affective judgments[8–10].

Spontaneous facial mimicry may play an instrumental role in shaping our interpretation of expressions[11]. Indeed, the movement of facial muscles provides afferent proprioceptive/somatosensory feedback to the brain[12–14], feedback that could assist in resolving visual ambiguities. In support of this idea, a number of studies demonstrate that blocking or interfering with facial mimicry in neurotypical participants[15], and in those with congenital facial paralysis[16], is associated with reduced emotion recognition ability. However, empirical evidence is mixed, and it remains controversial what role facial

mimicry, and its accompanying changes in facial feedback, play in emotion recognition[17].

A likely reason for at least some of the inconsistencies in the literature is that past studies investigating the issue were limited in their ability to precisely control which muscles were activated or inhibited to what degree. More importantly, the methods used by scholars to study the effects of altering facial mimicry and facial feedback offer only poor temporal control, and therefore cannot isolate effects on emotion recognition from other possible confounding factors, such as the effects of sustained effort of holding a pen between the lips (a commonly used technique to prevent facial mimicry). In order to overcome these limitations, we have recently started using computer-controlled facial neuromuscular electrical stimulation (fNMES) to activate specific facial muscles in a more controlled manner. For an overview of this method see[18].

[1]Department of Psychology, University of Essex, Colchester, UK. [2]School of Psychology, University of Derby, Derby, UK. [3]Centre for Clinical Brain Sciences, University of Edinburgh, Edinburgh, UK. [4]ArtScience Interfaculty, Royal Academy of Art, Royal Conservatory, The Hague, The Netherlands. [5]Department of Psychology, Webster Vienna Private University, Vienna, Austria. [6]Department of Cognition, Emotion and Methods in Psychology, University of Vienna, Vienna, Austria. ✉e-mail: joshua.baker@essex.ac.uk; sebastian.korb@essex.ac.uk

Using fNMES, we have demonstrated that bilateral activation of the *zygomaticus major* (ZM) muscle (the main muscle involved in smiling) increases the valence of self-reported emotion[19] and the likelihood of perceiving emotionally ambiguous facial expression as happy[20]. These findings support the notion that facial feedback contributes to felt and perceived emotion, and are in line with the facial feedback hypothesis[21,22]. Importantly, by using fNMES we were able to isolate the ZM muscles and precisely control the intensity, symmetry, onset and duration of their activation during the simultaneous presentation of facial expressions (i.e. for the duration of a presented face).

Current experimental methods of manipulating proprioceptive facial feedback (e.g. holding a pen between the lips) offers poor temporal control. That is, the induced feedback occurs prior to, and for the duration of a number of cognitive operations, and in some cases, over many trials. In contrast, one can use fNMES to manipulate facial feedback at much smaller timescales, which allows to investigate the chronology of visual and proprioceptive events during embodied emotion recognition. For example, one can target the period of spontaneous facial mimicry, which typically arises around 500 ms after a face is presented[23] or target periods of early visual processing. In doing so, one can elucidate the relative influence of facial feedback occurring at different times relative to the onset of a visual stimulus.

fNMES in combination with the high temporal resolution of EEG allows for the study of facial feedback effects at specific neural processing stages. We have previously demonstrated that fNMES artefacts can be removed from EEG, and accurate measurements of event-related potentials (ERPs) can be obtained[24,25]. Moreover, we showed that fNMES reduces the amplitude of a number of visual ERPs in response to facial expressions[20], although these reductions were not specific to a particular emotion. An ERP of particular interest when investigating embodied emotion recognition in faces is the N170. This is an early visual ERP related to the structural encoding of faces[26], often found to be larger for emotional relative to neutral faces[24,27–29]. Several studies have demonstrated that facial feedback contributes to emotion recognition and affects early visual processing (even before the typical onset time of spontaneous facial mimicry). For example, N170 amplitude for neutral faces becomes more similar to that for happy faces when participants pose a smile[30]. This suggests that facial feedback can indeed modulate early visual processing, and is not just instrumental from 500 ms after face onset onwards, when spontaneous facial mimicry typically occurs.

Emotional face perception likely relies on the combination of visual and somatosensory information. Somatosensory signals will influence perception more in cases where the visual signal is ambiguous. In line with this, Achaibou et al.[31] reported smaller N170 amplitudes in trials resulting in greater facial mimicry. Moreover, individuals with congenital facial palsy (with partial to extended facial muscle paralysis) demonstrate reduced connectivity (beta-band coherence) between somatosensory and visual areas during emotional face perception[32]. Coherence in the beta-band has been suggested to represent long-distance EEG connectivity for the processing of stimuli with affective value[33].

Modulations in N170 amplitude due to changes in facial feedback can also be contextualised within the framework of predictive coding[34]. The brain continuously generates predictions about sensory inputs and compares these with actual incoming sensory data[35]. If there is a mismatch, it results in a prediction error signal, which indicates that something unexpected has occurred. The internal model is then adjusted, so as to perpetually minimise these prediction errors. Indeed, N170 amplitude has previously been demonstrated to be sensitive to unexpected perceptual events[36], and the degree to which expectations are violated has a graded effect on N170 amplitude (the larger the violation, the larger the N170). Activations of ZM may therefore provide a certain context in which a happy face is predicted. If a different face, for example, is then presented (a violation of this prediction), then the resulting prediction error may manifest as an increase in N170 amplitude[30].

The current preregistered study (https://tinyurl.com/4wjy78nb) used fNMES in order to investigate whether facial feedback from ZM available during early visual processing (i.e. immediately prior to and during low level visual processing) or later visual processing (i.e. at the time spontaneous facial mimicry typically becomes measurable) has differential effects on the emotion categorisation (happy, sad) of neutral, happy, and sad facial expressions. EEG was recorded throughout to identify neural correlates of these effects. Prior to the main experiment, we ran two pilot studies (see Supplementary Materials) in order to select the best stimulus set, and to establish the onset time of spontaneous facial mimicry, to confirm the timing of late fNMES. In the main experiment, faces were presented for 1 s and fNMES was delivered at 70 Hz during *early* (−250 ms to 250 ms relative to face onset) or *late* (500 ms to 1000 ms relative to face onset) visual face processing, or not at all (*off* condition).

We expected (H1) that both early and late fNMES would significantly increase the likelihood of labelling neutral faces as happy, relative to no fNMES. We also expected (H2) this effect to be larger for the late fNMES period, which overlaps with the time period when spontaneous facial mimicry to emotional faces typically arises (as also confirmed in our EMG pilot experiment) and therefore should be the most natural timing for facial feedback changes to occur during emotion processing. Moreover, we expected (H3) that early fNMES would reduce the amplitude of the N170 in response to all faces, as was observed in our previous study[20] and that this reduction would predict how faces were eventually labelled. Finally, we performed exploratory analyses on beta-band coherence between somatomotor and visual areas of the brain, in order to identify potential neural correlates of the effects of late fNMES on choice.

## Methods

We have provided a full account of our sample size determination, justifications for data exclusion and comprehensive descriptions of all measures used within our research. The methods and hypotheses were registered in November 2023, before data acquisition started. The pre-registration, stimuli, tasks, analysis scripts and data are openly accessible at Open Science Framework (https://tinyurl.com/4wjy78nb). The analysis scripts are also available on Zenodo[37].

### Participants

We performed a power analysis for finding effects at the behavioural level (on neutral trials only) using data simulations in R. Drawing from the study by Efthimiou et al.[20], we assumed a small effect size (b = 0.09) for the early condition, indicating a 9% increase in the likelihood of classifying facial expressions as happy. For the late condition, we expected this effect to double in size (b = 0.18). Using the simr package in R, we ran for various sample sizes 1000 simulations of generalised linear mixed effects models using a binomial distribution and including a full random effects structure with all individual slopes and intercepts. A sample size of 45 participants was found to provide an average statistical power of 87% (95% CI = 78.80% to 92.89%) to detect a significant effect of fNMES in the early condition.

The participants were 51 adults (28 females, 23 males, mean age = 22.9, SD = 3.65, range 18–33), with normal or corrected to normal vision, no current use of prescribed medication or history of illicit drug use, and no history of neurological or psychiatric illness. Participants were recruited through a number of channels (e.g. flyers, email lists, social media) and were financially compensated at a rate of £10 per hour. They gave written informed consent before taking part, and were debriefed at the end of the testing session. The study was approved by the local ethics committee (ETH1920-0847). The analyses concerning only neutral expressions (that is, comparisons between neutral faces labelled as happy with those labelled as sad), was carried out in 48 participants, as three participants had labelled all neutral faces as belonging to only one category.

### Stimuli, task design and experimental procedure

The stimulus pool consisted of 48 greyscale images of computer-generated facial expressions from 16 different identities (8 female). Faces were generated with the FaceGen software (www.facegen.com), and their emotional expressions were created based on the Facial Action Coding System

(FACS)[38] using the FACSGen software[39,40]. This process involves manipulating Action Units (AUs), which correspond to specific muscle (group) movements that contribute to facial expressions. Each identity displayed a neutral, sad and happy expression, which were selected based on ratings from pilot study 1 (see Supplementary Materials). The expressions of happiness included AUs 6, 7 and 12, while sadness included AUs 1, 4, 7, 11 and 15. Each of the neutral expressions was repeated six times, and each happy and sad expression was repeated four times, in each fNMES condition (off, early, late; see below). This resulted in a total of 288 neutral faces, 192 happy faces and 192 sad faces being displayed to each participant. Five additional images were used for practice trials at the onset of the experiment. Scrambled faces (generated by scrambling the face area of the neutral face from each identity) were also presented immediately following each face on every trial to supress afterimages.

Participants completed six blocks of trials, each lasting ~6 min. Trials included neutral, happy, or sad faces during three different fNMES conditions (off, early, late). An additional 50 trials with fNMES but no face presentation were included in the task, but discarded from analyses, as a double-subtraction method was instead used to further remove fNMES noise in the EEG (see "Methods and Results"). More details on fNMES parameters are presented below. Participants completed 722 trials in total, which were presented in a pseudo-random order, whereby no more than four trials of the same fNMES type and no more than four trials of the same emotion type were presented consecutively. Importantly, two thirds of all trials included a 500 ms train of 70 Hz fNMES at motor threshold (MT) intensity to bilateral ZM (early and late conditions), with the other third of trials without stimulation (off condition). Specifically, in the early fNMES condition, stimulation started 250 ms prior to and was maintained until 250 following the onset of the face. In the late fNMES condition, an identical stimulation was delivered from 500 to 1000 ms after the onset of the face. The time window selected for the late fNMES conditions was based on the literature, as well as on pilot study 2 (see Supplementary Materials), where statistically significant facial mimicry to happy vs neutral faces was found from 500 ms onwards, using facial EMG.

The experiment was programmed in PsychoPy v2021.1.4[41]. Each trial (see Fig. 1) began with a centrally presented white fixation cross (horizontal = 0.47° and vertical = 0.47° of visual angle) on a black background for 1000 ms, which was then followed by a face stimulus for 1000 ms (horizontal = 14° and vertical = 20° of visual angle). Thereafter, a scrambled face was presented for a jittered duration of 200–300 ms, followed by a final screen displaying the words 'happy or sad?' that invited participants to provide a button press on a number keyboard (4 or 6, button assignment counterbalanced across participants) in order to label the previously presented face as either happy or sad.

Participants were first welcomed into the lab and were provided with consent forms and study information. Participants then provided their gender (choosing between the options: 'male', 'female', 'other', 'prefer not to say') and completed the PANAS questionnaire[42] in order to measure positive and negative affect prior to the experiment (t1). Information on participants' race/ethnicity was not collected. Head measurements were then taken before an appropriately sized EEG cap was fitted to the participant's head. Following the application of conductive gel (signaGel), participants sat in a comfortable chair in a sound-attenuated booth ~60 cm away from a 24.5 inch screen (Alienware aw2521h) with a resolution of 1920 × 1080 pixels and a 60 Hz refresh rate. Prior to the onset of the experiment, a calibration of the fNMES equipment was performed. The experimenter cleaned the skin of the participants' cheeks using 70% iso-propyl alcohol wipes. Two pairs of disposable electrodes were placed over the bilateral ZM muscles, following electromyography (EMG) guidelines[43,44]. To identify the best positioning of the electrodes and ensure that a weak smile could be induced comfortably, fNMES intensity was gradually increased until visible muscle contractions were observed (the motor threshold, or MT). Participants' faces were recorded throughout the testing session using a webcam mounted on top of the stimulus-displaying screen. This was to ensure the correct placement of stimulation electrodes,

and to record the intensity of muscle movement on each trial. Finally, on completion of all trials, participants again completed the PANAS questionnaire (t2).

## fNMES parameters and procedure

The delivery of fNMES to the bilateral ZM muscle was achieved using two constant-current electrical stimulators (Digitimer, DS5 https://tinyurl.com/yta3wa3a), each interfaced with a digital to analogue converter[25]. A 500-ms long train of biphasic square pulses (100-μs biphasic pulse width and 14-ms delay between biphasic pulses) was delivered at 70 Hz using disposable Ag/AgCl electrodes measuring 16 × 19 mm (Ambu BlueSensor BRS). Average stimulation intensity was 22.60 mA ($SD$ = 3.62, range: 14.25–33.75), and current density per electrode area ($M$ = 0.62, range 0.39–0.92) was well below the 2 RMS mA/cm2 safety threshold[18].

## EEG data acquisition and signal processing

EEG data were acquired with a waveguard cap containing 64 Ag/AgCl electrodes in the international 10–20 configuration, and an eEGo sports amplifier (ANT neuro, Netherlands), at 512 Hz and digitised with 24-bit resolution. Data were referenced online to electrode CPz, with the ground electrode at AFz. For further analyses, EEG data were imported and processed using functions from the EEGLAB (v2022.1) environment[45] for MATLAB (The Mathworks, Inc.). We followed a previously established procedure for the cleaning of fNMES-induced artefacts[24]. Continuous EEG were first filtered with a 0.5 Hz high pass and 80 Hz low pass filter, channels with excessive noise or artefacts were identified through visual inspection and interpolated, line noise was removed using Zapline and Cleanline, and data were epoched from 1000 ms before to 1200 ms after face stimulus onset. We performed independent component analysis (ICA) on the epoched data using the runica function in EEGLAB and removed components representing blinks and fNMES artefacts[24]. Following the removal of ICA components, trials were rejected if any values exceeded ±100 μV, resulting in an average of 72 trials per condition included in the final analysis. Baseline correction was then applied using the mean value between −500 and −260 ms pre-stimulus onset (i.e. a period of time not including fNMES). The data were finally filtered with a 40-Hz low pass filter and re-referenced to the common average.

## Analysis of behavioural data

Statistical analyses were conducted in R (R Core Team, 2020), implementing (generalised) mixed models with the lme4[46] package, p values were computed using the lmerTest package[47], approximations of Cohen's d for generalised mixed effects models were computed using the formula d = beta/(pi/sqrt(3)), as suggested by[48]. Quality of the statistical models (including distribution of residuals) was verified with the performance package[49]. The first of our pre-registered models tested that fNMES would increase the percentage of participants' choices of happiness in response to neutral faces. To this end, we used a Generalised Linear Mixed Effects model (GLMM) using a binomial distribution that included the categorical fixed effect of fNMES (off, early, late) and by-subject intercepts and slopes as random effects: Choice ~ fNMES + (fNMES | Participant). A second model included the additional fixed effect of emotion (neutral, happy, sad) to elucidate eventual interaction effects between fNMES and emotion on choice: Choice ~ emotion * fNMES + (1 | participant). Its random effects structure was simplified to prevent singular fits. Posthoc tests were carried out using the package emmeans[50] with Bonferroni correction. For key non-significant effects Bayes factors were computed with the BayesFactor package and default priors. In addition, we also fitted the above model including two covariates: PANAS difference scores (negative and positive affect scores at t2 minus t1) in order to examine the potential influence of mood changes on behaviour, and the Point of Subjective Equality (PSE) values. PSE values were derived for each participant for each fNMES condition and represented the point on the emotion continuum (sad coded as −1, neutral coded as 0 and happy coded as 1) that participants were equally likely to choose happy or sad.

Fig. 1 | **Trial structure and example stimuli. a** each trial began with a fixation cross which was followed by either a neutral, happy, or sad face, displayed for 1000 ms. Following a scrambled face, participants responded via keypress as to whether they had perceived the face as happy or sad. fNMES was either delivered in an early period (−250 to 250 ms around face onset), late period (500–1000 ms post face onset), or not at all (off condition). **b** example of a sad (left), neutral (middle) and happy (right) facial expression from a female identity. **c** schematic of electrode placement over the participant's bilateral zygomaticus major muscles.

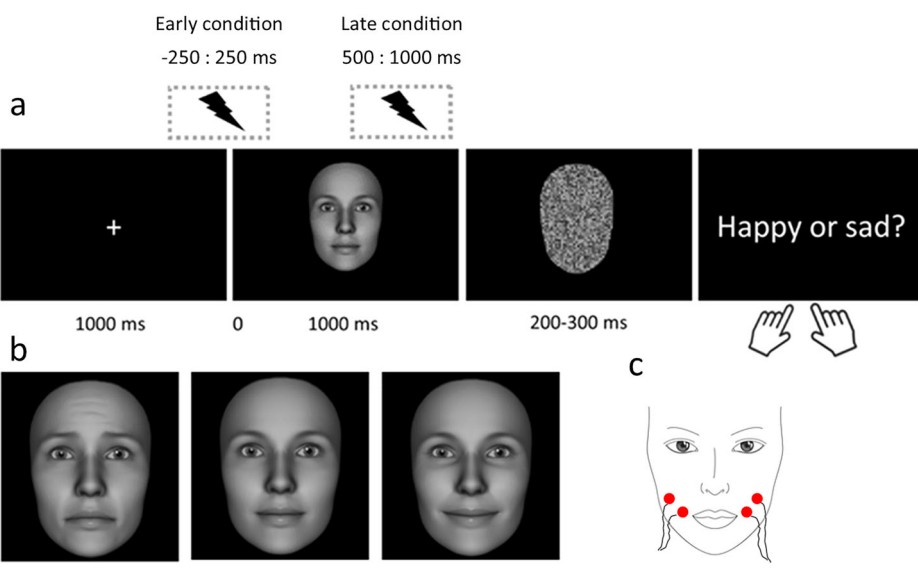

The degree of fNMES-induced smiling on each trial was captured with video recordings and estimated with automatic FACS coding[38] implemented in OpenFace[51]. Firstly, videos of each trial were extracted for 46 participants (6 participants were not included given poor video data), which included 2 s before and after stimulation onset, or the face onset in the case of the off condition. Following baseline correction (using the first second of each video), AU intensity over time (% of baseline values) was calculated for AUs 6, 12, 15 and 4 (see Fig. 2) in each of the experimental conditions. Trials in each condition were then averaged together, and the mean intensity value for AU12 was calculated in the stimulation period. A linear model with the predictors fNMES (off, early, late) and emotion (sad, happy, neutral) was then used to compare smile intensity (AU12) in each condition.

### Analysis of EEG

**Time domain analyses.** Pre-processed and epoched EEG data were first analysed in the time domain. A mass-univariate analysis approach[52] was utilised in order to identify regions in time and space that presented fNMES-induced significant differences between happy and sad faces. The mass univariate approach involves computing multiple paired-sample t-tests across each time point for each channel for two different waveforms. Specifically, the following difference waveforms were computed and entered into the analysis. First, we subtracted sad face trials from happy face trials, in each of the three fNMES conditions. Following this first differencing procedure, we then directly compared these differences between the off and early fNMES conditions, and separately, the off and late fNMES conditions. The resulting derived waveforms therefore expressed the effects of fNMES on the difference between happy and sad faces. In using this approach, one can avoid directly comparing trials that contain fNMES with those that do not. Such differencing procedures can eliminate fNMES artefacts over and above our established procedure[24].

Visual inspection of the computed p-value maps (with false discovery rate correction) derived from comparing fNMES off and early conditions only revealed a significant bilateral occipito-temporal cluster from 150 to 200 ms post face onset, which we interpreted as the N170 component based on timing and topography. We derived N170 amplitude in each trial by averaging the identified significant channels (P5, P3, P1, Pz, P2, P4, P6, P8, PO7, PO5, PO3, POz, PO4, PO6, PO8, O1, Oz, O2) and deriving the mean value between 150 and 200 ms post face onset. A similar approach by comparing trials in the fNMES off and late condition revealed no significant differences (even at more liberal significance thresholds).

To examine whether N170 amplitude to neutral faces predicted how they were labelled in the early compared to the off condition, we used a GLMM with a binomial distribution with the categorical predictor fNMES (off, early), and the continuous predictor N170: Choice ~ fNMES * N170 + (fNMES * N170 | participant). A second model that included all face emotions was also fitted.

**Coherence analyses.** In addition to analysing N170 amplitude, we performed an exploratory analysis that examined whether fNMES would modulate spectral coherence between somatomotor and occipital areas during trials in which neutral faces were labelled as happy relative to when they were labelled as sad. To this end, we derived time-frequency resolved coherence coefficients between central and occipital regions of interest (ROIs) for neutral faces labelled as happy or sad for each fNMES condition. First, we derived coherence estimates between 0 and 1 s post-stimulus and between 5 and 40 Hz from the power and cross-spectra using a frequency resolution of 0.5 Hz, a time resolution of 10 ms and, for each given frequency, hanning windows with a length corresponding to 5 cycles between a select number of electrode pairs. Specifically, each electrode in the central ROI (Cz, C1, C2, C3, C4) was paired with each electrode in three separate occipital regions (left: P1, P3, P5, P7, PO3, PO5, PO7, middle: Pz, POz, Oz and right: P2, P4, P6, P8, PO4, PO6 and PO8). We then averaged the resulting coherence values between central and left occipital electrodes, central and middle occipital electrodes and central and right occipital electrodes, resulting in three coherence maps for each fNMES condition (off, early, late) for neutral faces labelled as happy and those labelled as sad.

Afterwards, we statistically assessed the categorisation choice (happy or sad) by fNMES interaction by performing a Monte Carlo based cluster permutation test based on a dependent samples t-test (two-tailed, 2000 permutations, cluster alpha = 0.01 and final alpha = 0.05) separately for each ROI on the difference in coherence between neutral faces rated as happy and sad in the early fNMES condition, compared to the difference for neutral faces labelled as happy and sad in the off condition. This was also performed to contrast the late fNMES and off conditions. This resulted in two clusters that showed that both early and late fNMES modulated the difference in coherence between neutral faces labelled as happy vs. labelled as sad. Both of these identified clusters expressed fNMES-induced changes in beta-band coherence from 650 ms post stimulus onset between the central and left-occipital region. Finally, both significant clusters were integrated, which generated a mask that included coherence values for frequencies and time points that expressed significant differences in the early-off and late-off contrasts, in order to extract mean coherence measures in each condition separately.

To first test whether coherence differed between neutral faces labelled as happy vs. sad in each of the fNMES conditions, we implemented a 3

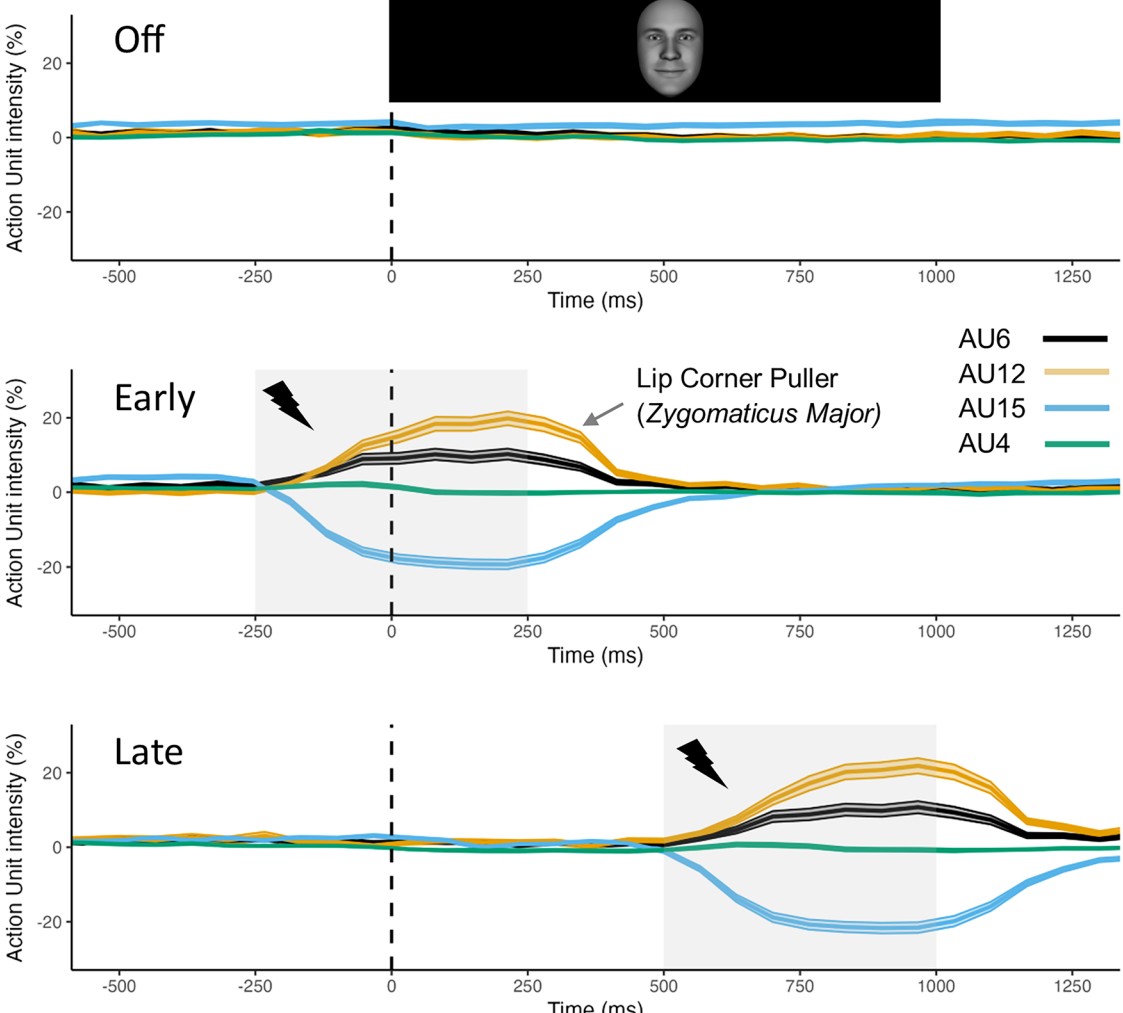

**Fig. 2 | Automatic FACS coding.** Action unit intensity for AU6 (Orbicularis Oculi, i.e. Cheek Raiser), AU12 (Lip corner puller, i.e. Zygomaticus Major), AU15 (Lip corner depressor, i.e. Depressor Anguli Oris), and AU4 (Brow lowerer, i.e. Corrugator Supercilii) over time in the off (top), early (middle) and late (bottom) fNMES conditions. In both the early and late fNMES conditions and during the period of stimulation (grey boxes), AU12 intensity can be seen to increase relative to the off condition. Dotted line indicates the onset time for face stimuli. Shaded areas show 95% CI. N = 46.

(fNMES: off, early, late) x 2 (choice: happy, sad) repeated measures ANOVA. Posthoc analyses were performed using Bonferroni corrected paired-sample t-tests. In order to test whether coherence during trials containing neutral faces was predicted by fNMES condition, we used a linear model with the fixed factor fNMES (off, early, late), with off serving as the reference level. As such, the model formula was: Coherence ~ fNMES. Given that coherence estimates are not derived at the single-trial level (e.g. based on averages across conditions), the investigation of the relationship between coherence and choice was performed at the condition-average level. As such, we computed an average behavioural measure (percentage of neutral trials labelled as happy) in each fNMES condition to include in the following models. First, we fitted a linear model that predicted percent choice happy for neutral faces by coherence. Second, we fitted a model that also included fNMES (off, early, late) as a categorical predictor (percent choice happy ~ fNMES + coherence), with the fNMES off condition serving as the reference level.

## Results

### fNMES induces smiling as confirmed by automatic FACS coding

In order to confirm that fNMES activated AU12 (Lip Corner Puller; *Zygomaticus Major*), we performed automatic FACS coding on video footage captured with a webcam (see Fig. 2). Mean intensity values

(% change relative to baseline) were computed for AU12 during the stimulation period in each condition. We implemented a linear model with the predictors fNMES (off, early, late) and emotion (happy, sad, neutral), which revealed significant differences between fNMES early ($M = 0.137$, $SE = 0.01$) vs. off ($M = 0.002$, $SE = 0.01$) [$b = 0.12$, 95% CI (0.06, 0.18), $SE = 0.03$, $p < 0.001$] and between fNMES late ($M = 0.156$, $SE = 0.01$) vs. off [$b = 0.15$, 95% CI (0.09, 0.21), $SE = 0.03$, $p < 0.001$], while fNMES early and late did not differ from each other [$b = 0.02$, 95% CI (−0.03, 0.08), $SE = 0.03$, $p = 0.42$]. Converting this to a type 3 anova model using the car package revealed a medium sized main effect of fNMES ($\eta p2 = 0.06$). To summarise, fNMES successfully increased AU12 intensity during the stimulation periods, confirming a smiling expression.

### fNMES increases the likelihood of labelling neutral faces as happy, regardless of the timing

Our first pre-registered analysis concerned only neutral faces and was to predict choice (happy or sad) by fNMES condition (off, early, late), with off serving as the reference level (conditional $R^2 = 0.36$, marginal $R^2 = 0.001$). A significant main effect of early fNMES was found [$b = 0.17$, $z = 3.84$, 95% CI (0.09, 0.27), $SE = 0.046$, $p < 0.001$, $d = 0.10$], as was a significant main effect of late fNMES [$b = 0.133$, $z = 2.88$,

**Fig. 3 | The effects of fNMES on choice. a** Box plots showing the percentage of choosing happy for happy (left) and sad (right) faces in each fNMES condition. Regardless of the fNMES condition, participants had high accuracy and consistently labelled happy faces as being happy, and sad faces as being sad. **b** Box plot (left) and probability density plot (right) showing the percentage and probability of choosing happy for neutral faces in each fNMES condition. Both early and late fNMES conditions significantly increased the likelihood of choosing happy for neutral faces. Thick black lines in the box plot show condition means with coloured boxes showing standard deviation. The grey line spanning the plot shows the mean in the off condition. Vertical lines in the probability density plot show mean probability of choosing happy in each fNMES condition for neutral faces. ** $p = 0.002$, * $p = 0.032$. N = 51.

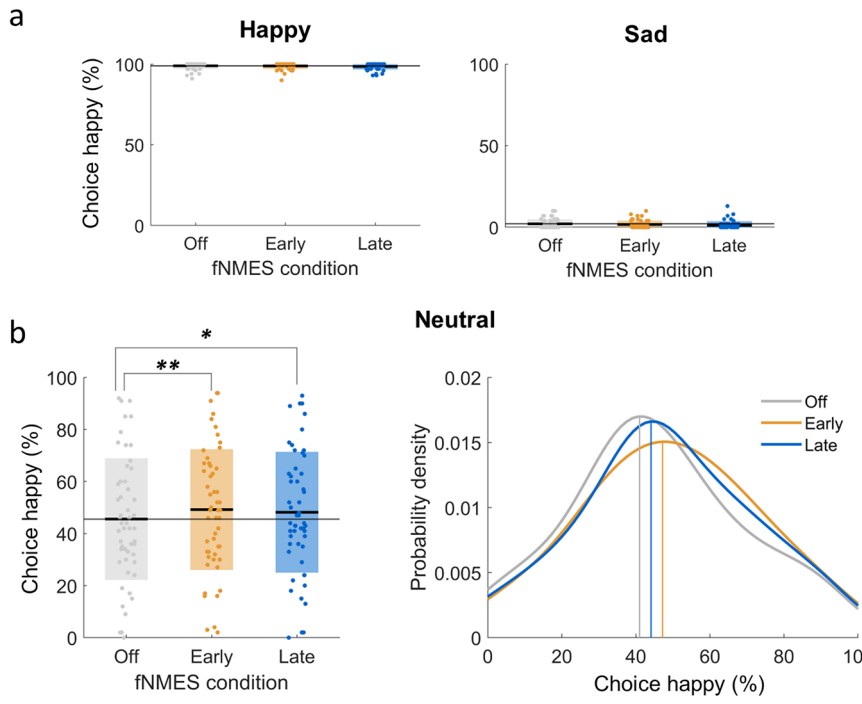

95% CI (0.04, 0.22), SE = 0.04, $p = 0.003$, $d = 0.07$], indicating, as predicted, that neutral faces were more likely to be categorised as happy when fNMES was delivered to ZM, regardless of the timing of fNMES (Fig. 3b). We had hypothesised that fNMES delivered in the late period (i.e. during the natural timing of spontaneous facial mimicry) would have a greater effect than when it was delivered in the early period, however, posthoc tests revealed that this was not the case [$b = 0.04$, 95% CI (−0.07, 0.15), $z = 0.96$, SE = 0.05, $p = 1.00$]. A Bayesian paired t-test with a standard Cauchy prior revealed moderate evidence in favour of the null hypothesis ($BF_{10} = 0.27$), suggesting no significant difference between early and late effects on neutral faces.

In order to explore if fNMES effects were specific to neutral faces, a second model included all emotions (happy, sad, neutral, with neutral faces serving as the reference level). This model (conditional $R^2 = 0.81$, marginal $R^2 = 0.74$) revealed expected significant main effects of happy faces [$b = 5.50$, $z = 29.22$, 95% CI (5.14, 5.87), SE = 0.18, $p < 0.001$, $d = 3.03$] and sad faces [$b = −4.23$, $z = 29.72$, 95% CI (−4.52, −3.96), SE = 0.14, $p < 0.001$, $d = −2.34$], and significant main effects of both early and late fNMES (which did not differ from each other, as identified in the previous model and the Bayesian t-test). Posthoc tests that examined the effects of early and late fNMES for each emotion indicated that the impact of fNMES on choice was significant for neutral faces [Early fNMES: $b = −0.17$, 95% CI (−0.28, −0.06), $z = −3.78$, SE = 0.04, $p < 0.001$, Late fNMES: $b = −0.13$, 95% CI (−0.23, −0.02), $z = −2.84$, SE = 0.04, $p = 0.013$], but was not significant for happy faces [Early fNMES: $b = 0.26$, 95% CI (−0.31, 0.83), $z = 1.07$, SE = 0.24, $p = 0.84$], Late fNMES: $b = 0.52$, 95% CI (−0.02, 1.06), $z = 2.31$, SE = 0.22, $p = 0.06$] or sad faces [Early fNMES: $b = 0.16$, 95% CI (−0.32, 0.65), $z = 0.81$, SE = 0.20, $p = 1.00$], Late fNMES: $b = 0.41$, 95% CI (−0.10, 0.93), $z = 1.93$, SE = 0.21, $p = 0.16$]. The pattern of results remained the same when including as covariates the PANAS difference scores or the PSEs (although the model with the PSEs resulted in singular fits and posthoc tests no longer showed a significant difference between fNMES conditions for neutral faces, probably because of the overlap between the fNMES and PSE variables).

To summarise, fNMES to bilateral ZM in both the early and late time windows increased the likelihood of labelling neutral faces as happy to a similar degree.

### fNMES decreases N170 amplitude and moderates the relationship between N170 and choice

Our second preregistered analysis was to examine whether N170 amplitude in interaction with fNMES predicted choice for neutral faces. Electrodes and time points to be included in this analysis were selected based on a mass-univariate analysis (Fig. 4, see 'Methods' section for more details).

We used a model that included the categorical predictor fNMES (off, early, with off serving as the reference level) and the continuous predictor N170 (marginal $R^2 = 0.003$). A main effect of fNMES was observed [$b = 0.14$, $z = 2.42$, 95% CI (0.03, 0.27), SE = 0.06, $p = 0.01$, $d = 0.08$], indicating again that early fNMES increases the likelihood of labelling neutral faces as happy. Importantly, a significant interaction between fNMES and N170 amplitude was also observed [$b = 0.02$, $z = 1.99$, 95% CI (0.01, 0.04), SE = 0.01, $p = 0.04$, $d = 0.01$]. In the off condition, a negative slope (−0.01, SE = 0.01, $z = −1.44$, $p = 0.15$) revealed that a larger N170 amplitude (more negative value) increased the likelihood of labelling a neutral face as happy (see Fig. 5). In contrast, during early fNMES, the opposite was observed, in which a positive slope (0.01, SE = 0.01, $z = 1.18$, $p = 0.24$) revealed that a smaller N170 (more positive value) increased this likelihood. By itself, neither slope significantly differed from zero, as shown by simple slope analyses. In order to assess whether the same pattern was present for all faces (i.e. if N170 in interaction with fNMES predicts choice of all faces), we utilised the same model including all trials in the early and off conditions (marginal $R^2 = 0.002$). This again revealed a significant interaction between fNMES and N170 [$b = 0.02$, $z = 4.91$, 95% CI (0.02, 0.04), SE = 0.01, $p < 0.001$, $d = 0.01$] that showed the same pattern as when analysing only neutral faces. A simple slope analysis revealed that both the negative slope in the off condition [$b = -0.02$, $z = −3.72$, SE = 0.01, $p < 0.001$] and the positive slope in the early condition [$b = 0.01$, $z = 2.89$, SE = 0.01, $p < 0.001$] were significantly different from zero.

To summarise, early fNMES is associated with a decrease in N170 amplitude, which is linked to an increased probability of labelling faces as happy (Fig. 5).

### fNMES modulates beta-band coherence for neutral faces

Cluster based permutation tests resulted in two clusters showing significant fNMES-induced modulations of beta-band (13–22 Hz) coherence between central and left occipital regions from 650 ms post stimulus onset (see

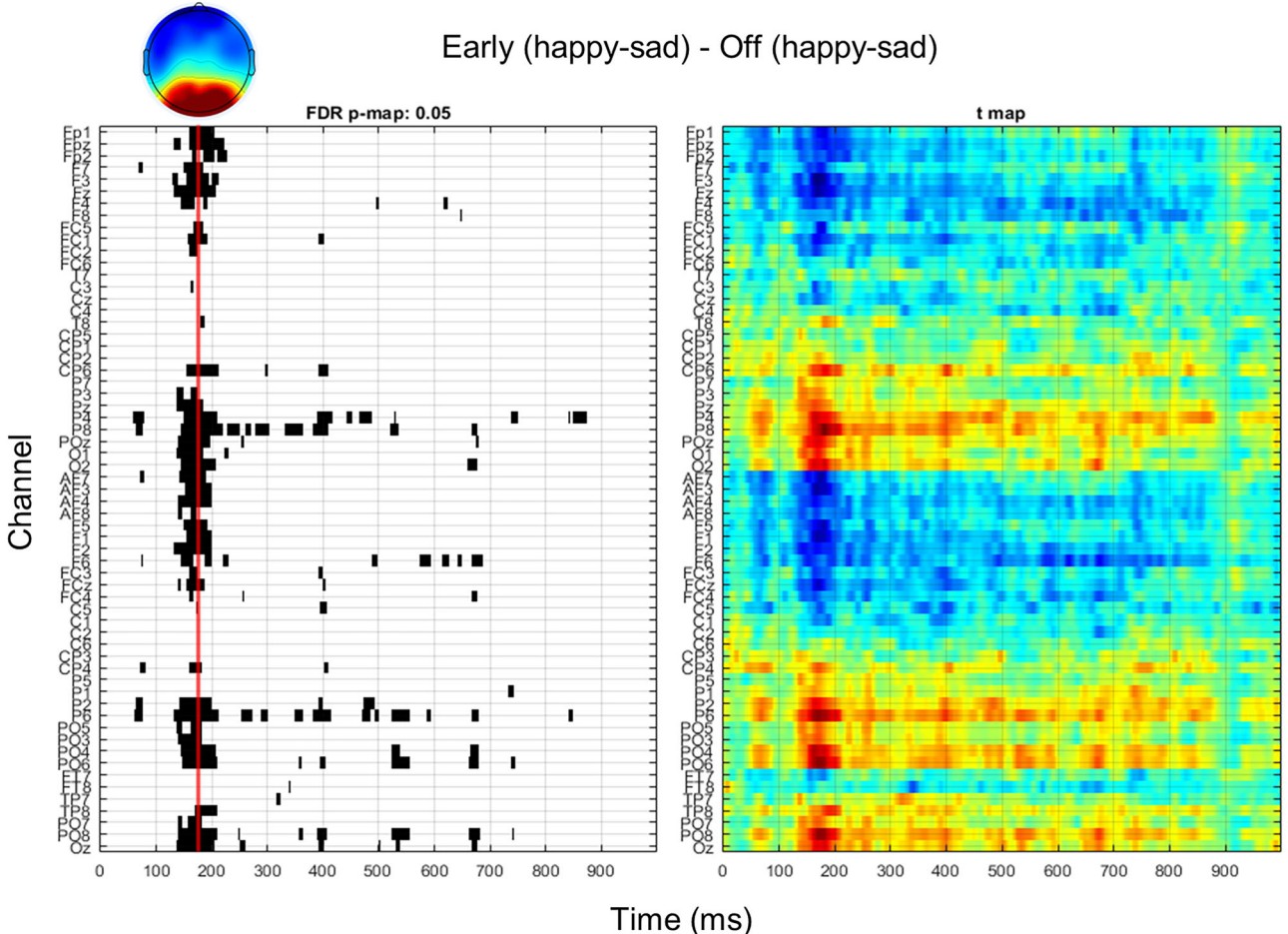

**Fig. 4 | Mass-univariate analysis were used to derive effects of fNMES on the difference between happy and sad faces (impacting the N170).** Sad trials were first subtracted from happy trials in the fNMES early condition, which was then compared to the same difference in the off condition. p-value map (left) with false-discovery rate correction showing significant effects of early fNMES on happy-sad differences, with the corresponding t-values (right). Topographic map showing the voltage difference for the same contrast between 150 and 200 ms post face onset. $N = 51$.

Fig. 6a). We derived a mask that incorporated both of these clusters in order to derive mean coherence estimates for neutral faces labelled as happy and those labelled as sad, in each fNMES condition.

In order to test whether beta-band coherence differed between neutral faces labelled as happy vs. those labelled as sad in each fNMES condition, we first implemented a 3 (fNMES: off, early, late) x 2 (choice: happy, sad) repeated measures ANOVA. There were no significant main effects of fNMES [$F (1.91, 89.59) = 1.34$, $p = 0.267$, ηp2 = 0.028, 95% CI (0, 1)] or choice, [$F (1, 47) = 0.12$, $p = 0.735$, ηp2 = 0.002, 95% CI (0, 1)], but a significant interaction between fNMES and choice [$F (1.92, 90.45) = 19.58$, $p < 0.001$, ηp2 = 0.294, 95% CI (0.17, 1)] was observed. Posthoc tests revealed that in the off condition, beta-band coherence was greater for neutral faces labelled as happy ($M = 0.340$, $SE = 0.016$), than those labelled as sad ($M = 0.316$, $SE = 0.016$) [$t (47) = 2.95$, 95% CI (0.008, 0.04), $p = 0.004$]. In contrast, in the late fNMES condition, coherence was greater for those labelled as sad ($M = 0.332$, $SE = 0.016$) than as happy ($M = 0.318$, $SE = 0.015$) [$t (47) = -2.24$, 95% CI ($-0.02$, $-0.001$), $p = 0.029$]. A similar pattern was observed in the early fNMES condition, however this was not significant [$t (47) = -1.84$, 95% CI ($-0.03$, 0.001), $p = 0.072$].

In order to test whether beta-band coherence in interaction with fNMES predicted choice for neutral faces, we used a model that included the categorical predictor fNMES (off, early, with off serving as the reference level) and the continuous predictor coherence. No main effects or interactions were observed (Early fNMES: $b = 3.80$, $SE = 14.06$, $t = 0.27$, 95% CI ($-23.9$, 31.5), $p = 0.78$; Late fNMES: $b = 3.31$, $SE = 13.99$, $t = 0.24$, 95% CI ($-24.3$, 30.97), $p = 0.81$; Beta Coherence: $b = 7.83$, $SE = 28.71$, $t = 0.23$, 95% CI ($-48.9$, 64.6),

$p = 0.79$; Early fNMES x Beta Coherence: $b = -0.27$, $SE = 41.18$, $t = -0.006$, 95% CI ($-81.6$, 81.1), $p = 0.99$; Late fNMES x Beta Coherence: $b = -1.50$, $SE = 40.70$, $t = -0.04$, 95% CI ($-81.9$, 78.9), $p = 0.97$).

To summarise, although fNMES had differential effects on beta-band coherence for neutral faces eventually labelled as happy vs. sad, these fNMES-induced modulations did not predict the emotion categorisation of neutral faces.

## Discussion

The aim of the current study was to examine the temporal dynamics of facial feedback effects in facial emotion recognition. Traditional manipulations of facial muscles afford very little temporal precision, whereby engagement/ blocking of facial muscles persists through all stages of face processing. As such, it is unclear whether changes in facial feedback during early visual perception or relatively later stages (e.g. when spontaneous facial mimicry typically occurs) have differential effects on emotional face recognition. To this end, we manipulated facial feedback from the ZM muscle using fNMES in an early ($-250$ to $+250$ ms) and a late time window ($+500$ to $+1000$ ms) relative to the onset of neutral, happy and sad facial expressions. We examined how categorisation choice (happy or sad) for each face would be modulated by fNMES. In addition, we examined the neural correlates of the effects of fNMES on choice by identifying an emotion x fNMES effect on the N170 in a data-driven way (mass univariate analysis), and by computing beta-band coherence between somatomotor and occipital areas of the brain.

Our findings demonstrate, as predicted (H1), an induced bias for categorising neutral faces as happy when receiving stimulation to the

**Fig. 5 | Relationship between N170 amplitude and choice. a** N170 for all faces categorised as happy (green) and sad (red) in the off (left) and early (right) fNMES conditions. Waveforms and topographic maps show the average of channels identified in the mass-univariate analysis, between 150 and 200 ms after face onset. Shaded areas show standard error. **b** Derived marginal means from a model predicting choice of each trial by fNMES and N170 amplitude in the off (left) and early (right) fNMES conditions. In the off condition, a larger N170 amplitude increased the likelihood of labelling a face as happy. In contrast, in the early fNMES condition, a smaller N170 increased this likelihood. Shaded areas show 95% confidence intervals. N = 51.

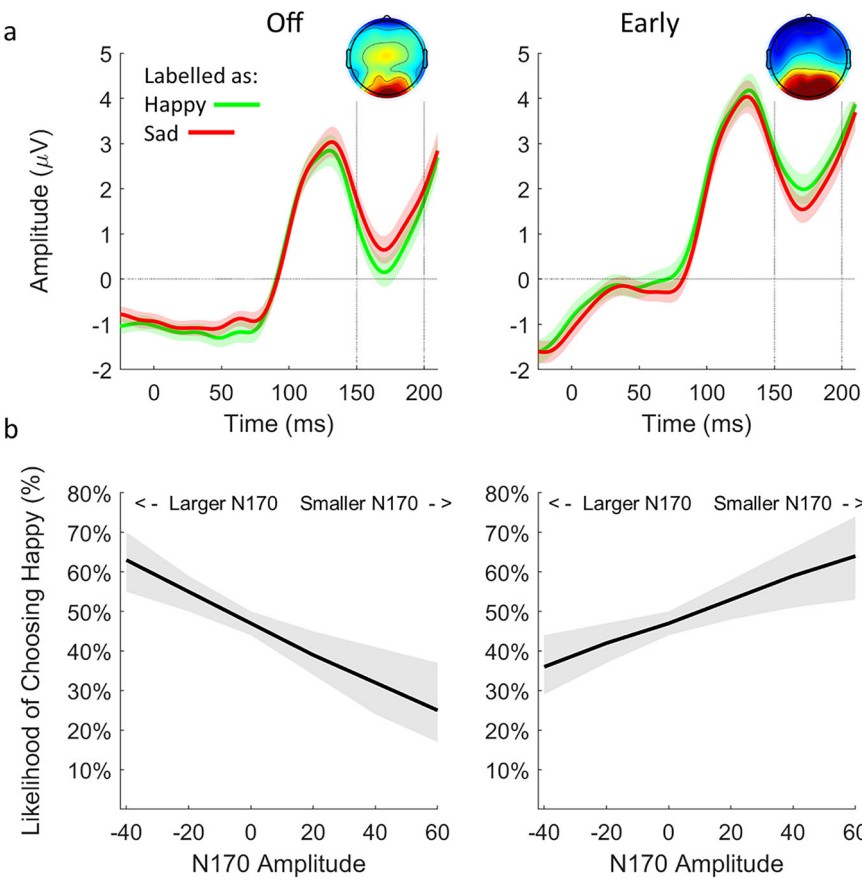

bilateral ZM muscle. This is consistent with our previous study[20] and provides clear evidence for the facial feedback hypothesis. It is important to emphasise, that fNMES modulated only the categorisation of neutral faces, and not of happy and sad faces. This was expected, given that multisensory integration is more likely to occur when stimuli in one or both sensory modalities are degraded or otherwise less informative[53]. In other words, in the absence of emotional information in neutral faces, participants tended to rely more on proprioceptive input. A similar pattern had been observed in our previous study that also showed a happiness bias in trials with fNMES to bilateral ZM muscles[20], although there the emotion by fNMES interaction did not reach significance, probably due to the fact that the stimulus set included only emotional expressions that were all relatively ambiguous.

We had also expected (H2) that the facial feedback effect on categorisation choices would be more pronounced when fNMES was delivered in the late period. The reason is that the nervous system might be particularly sensitive to changes in facial feedback during the period in which facial mimicry typically occurs (500 ms and onwards following the presentation of a face). Facial mimicry is considered to provide additional (proprioceptive) information in resolving visual ambiguities during emotional face recognition[11], and as such, changes in facial feedback accompanying facial mimicry do usually occur after the initial stages of visual face processing. However, in this experiment no differential effects of the timing of fNMES on categorisation choice were found. The reason we did not observe stronger effects in the later condition could be due to the fact that stimulation was not delivered in the optimal period (see "Limitations").

The observation that stimulation delivered in both time periods (immediately prior to and during initial stages of visual face processing, and relatively later during the expected period of spontaneous facial mimicry) resulted in the same induced response bias, suggests that fNMES is modulating distinct neural operations. This seems also supported by the different neural effects of early and late fNMES. Regarding stimulation in the early

period, we found that fNMES to ZM specifically reduced N170 amplitude. This is consistent with our previous study[20], however, in the current study we found a pronounced reduction that was specific to both actual happy faces, and neutral faces categorised as happy (importantly, categorisation responses always occurred at the end of each trial). Moreover, the larger the fNMES-induced decrease of the N170, the more likely a face was to be categorised as happy. This suggests that the presence of facial feedback from the ZM muscle during early visual perception (e.g. prior to N170) modulates the succeeding cortical excitability of the visual system. That is, visual processing is 'tuned down', given that somatosensory information from smiling muscles informs the brain that the presented face is happy. As a result, the reduction in N170 (and the degree to which this reduction predicts categorisation choice), suggests that early visual face processes involved in unpacking the emotional content of the face (such as the N170), operate to a lesser degree in the context of salient facial feedback information. In other words, the workload of the visual system is offset due to convincing proprioceptive evidence that a perceived face is happy. The larger this offset, the more likely participants categorise faces as happy. This finding is consistent with a study that demonstrated a relationship between the magnitude of facial mimicry, recorded with EMG over the ZM and corrugator supercilii muscles, and the amplitude of the preceding N170[31]. Trials that had smaller N170 amplitudes in response to faces were associated with more facial mimicry. As such, when visual processing was operating to a lower degree, more facial mimicry was needed to decipher the content of the face. We suggest that our findings present a similar trade-off between somatosensory and visual inputs.

An alternative explanation for this modulation in N170 is provided by predictive coding, a theoretical framework aiming to explain how the brain processes sensory information[34]. Stimulation of ZM may lead to the prediction that a happy face is to be presented, and a prediction error would occur if the presented face violates this prediction. Indeed, during early

**Fig. 6 | Beta coherence between central and left occipital region in response to neutral faces.**
**a** Coherence maps showing coherence for neutral faces labelled as happy vs. sad in the off condition (left) and for early-off (middle) and late-off (right) contrasts. Clusters (values encased in green boundaries) show significant differences in beta-band coherence from 650 ms post stimulus onset identified from the contrasts. **b** Mean coherence values for neutral faces labelled as happy (green) and sad (red) in each fNMES condition. In the off condition, neutral faces labelled as happy presented greater coherence than those labelled as sad. During late fNMES, however, the reverse was observed. A similar trend was also present for early fNMES. Error bars show standard error. p values are Bonferroni corrected. ** p < 0.01, *p < 0.05, o trend (p = 0.072). N = 48.

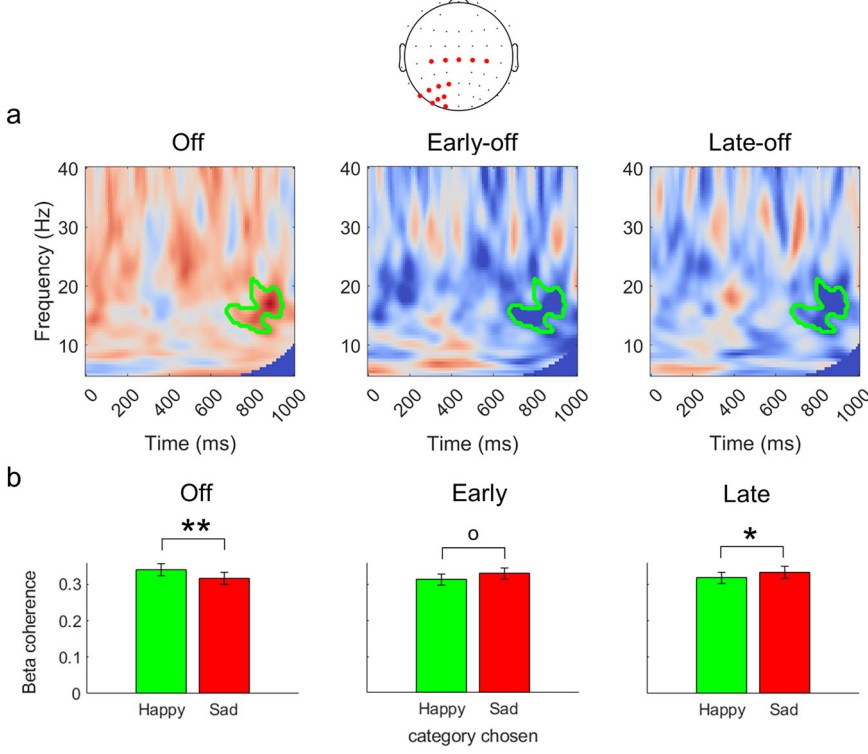

fNMES, N170 was greater (more negative) to sad than happy faces (see Fig. S3 in Supplementary Materials). As such, the unexpected occurrence of a sad face (given the activation of the participant's smiling muscles) resulted in a larger prediction error, reflected in a larger N170. Indeed, N170 amplitude has previously been demonstrated to be sensitive to unexpected perceptual events[36], and the degree to which expectations are violated has a graded effect on N170 amplitude (the larger the violation, the larger the N170). When prediction errors are small or absent, the N170 amplitude will become relatively smaller, which is what we found in response to happy faces (and neutral faces labelled as happy) during fNMES to the ZM muscles (the perceived face was consistent with the provided facial feedback context).

In contrast, the categorisation response bias induced with fNMES in the late period cannot be explained by the aforementioned modulations in N170 amplitude. Indeed, in the late condition stimulation was delivered from 500 ms following the onset of the face, which is well after the period of the N170. As such, we suggest that modulation of different neural mechanisms resulted in the same induced bias across the early and late fNMES conditions. However, a mass univariate analysis contrasting the late and off fNMES conditions did not reveal any significant differences in the time domain. As such, we considered modulations in functional connectivity between somatomotor and visual areas as a potential neural marker of the induced bias due to fNMES delivered in the late period. Beta-band coherence, for example, has previously been shown to be attenuated in individuals with congenital facial paralysis[32], potentially demonstrating limited information flow between somatomotor and visual areas during face processing. Coherence in the beta-band has been suggested to represent long-distance EEG connectivity for the processing of stimuli with affective value[33]. Our analysis revealed a modulation of beta-band coherence differences (between neutral faces labelled as happy vs. sad) from 650 ms onwards in the late fNMES conditions, relative to the off condition, in the left hemisphere. Specifically, beta-band coherence was greater for neutral faces labelled as happy than as sad in the off condition, however this relationship was reversed (i.e. greater for neutral faces labelled as sad) during late fNMES (with a similar non-significant trend observed in the early fNMES

condition). These findings were surprising, as one might expect that coherence for neutral faces eventually labelled as happy should be greater during fNMES, indicating an increase in information flow between somatomotor and visual areas (given the magnitude of changes in facial feedback). Our findings are opposite to this expectation, whereby a decrease (relative to neutral faces labelled as sad) was observed. Importantly, however, this modulation of beta-band coherence did not predict how neutral faces were eventually labelled (possibly due to lower statistical power because analyses could no longer be carried out at the single trial level). As such, despite the effects of fNMES on beta-band coherence, such modulations did not influence choice in our study. Although our analyses identified fNMES effects specifically in the beta-band, future studies might wish to consider exploring modulations of coherence in other frequencies during facial feedback manipulations. For example, gamma band activity has been found to represent the binding of stimulus features with top-down information[54].

## Limitations

We based our late stimulation period on a pilot study which only considered spontaneous facial mimicry to happy faces. As such, our late period might have been slightly ill-informed (i.e. did not reflect the timing of facial mimicry in the case of sad faces). Future studies should refine the late fNMES timing by testing alternative time windows (e.g. 400–800 ms, 600–1100 ms). Another possibility is that spontaneous facial mimicry manifests at a subthreshold level prior to being observable in facial EMG data. As such, although in a statistical sense facial mimicry occurred only after 500 ms following face onset, it is possible that undetectable state-changes in facial muscles already occur prior to this, and thus our late stimulation period was indeed too late. Future studies could investigate this further by providing fNMES, for example, from 200 ms onwards. The neural mechanisms underpinning the effects of late fNMES on choice requires further investigation as our analyses did not identify conclusive fNMES-induced modulations for this condition. Future studies might wish to focus solely on this time period, as only one third of the trials in the present study involved late stimulation. In doing so, one might achieve sufficient power to elucidate the neural correlates the effects of late fNMES

on choice. It is also important to consider that avatar faces, although allowing for tight control of low-level visual features, might differ from real pictures of facial expressions in their eliciting of processes involved in emotional recognition. As such, future studies might wish to present real-life, even dynamic facial expressions so as to maximise the generalisability of our findings.

## Conclusions

To conclude, the present study demonstrates that controlled changes in facial feedback from the smiling muscles either during early or relatively late visual processing can induce a happiness bias when labelling neutral facial expressions. Despite predicting that facial feedback effects would be more pronounced in a relatively late time window (during the period of facial mimicry), we found that pro-prioceptive information from facial muscles can exert an influence on behaviour both during this period, and to the same extent, during early visual processing stages. Notably, the observed bias introduced when delivering stimulation prior to and during early face processing can be explained by a reduction in N170 amplitude. We argue that the presence of salient somatosensory information from smiling muscles prior to the onset of a face relieved the visual system's load in deciphering the emotional content of that face. Indeed, the larger this reduction, the more likely a face was labelled as happy. Late fNMES induced a modulation of beta band coherence between somatomotor and occipital areas, which however was not predictive of categorisation choices. As such, the neural correlates of the effects of late fNMES remain poorly understood, and should be addressed in future studies specifically focusing on that period. Finally, it would also be interesting to explore the effects of facial muscle activation, and other controlled somatosensory inputs, on the perception of emotion in non-visual stimuli—for example, Selosse et al.[55] recently reported improved emotion recognition of fearful and angry voices, and modulation of early ERPS, when vocal cords were vibrated at emotion-congruent frequencies.

## Data availability

All the data are openly accessible at Open Science Framework (https://tinyurl.com/4wjy78nb).

## Code availability

All the code to run the experiment, preprocess and analyse the data are openly accessible at Open Science Framework (https://tinyurl.com/4wjy78nb).

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

## Acknowledgements

This work was funded by a Stand Alone Grant by the Austrian Science Fund, awarded to SK, MM and AE (Grant number: P 32637-B). The funders had no role in study design, data collection and analysis, decision to publish or preparation of the manuscript.

## Author contributions

Joshua Baker: Conceptualisation, Data Curation, Formal Analysis, Investigation, Software, Visualisation, Writing—Original Draft Preparation. Hong-Viet Ngo-Dehning: Data Curation, Formal Analysis, Supervision, Writing—Review & Editing. Themis N. Efthimiou: Conceptualisation, Data Curation, Formal Analysis, Writing—Review & Editing. Arthur Elsenaar: Conceptualisation, Funding Acquisition, Writing—Review & Editing. Marc Mehu: Conceptualisation, Funding Acquisition, Writing—Review & Editing. Sebastian Korb: Conceptualisation, Formal Analysis, Funding Acquisition, Supervision, Writing—Review & Editing.

## Competing interests

The authors declare no competing interests.
