## [Transparent Peer Review file · Communications Psychology]

Electrical stimulation of smiling muscles reduces visual processing load and enhances happiness perception in neutral faces

Corresponding Author: Dr Joshua Baker

Version 0:

Decision Letter:

Dear Dr Baker,

Thank you for your patience during the peer-review process. Your manuscript titled "The face says it all: electrical stimulation of smiling muscles reduces visual processing load and enhances happiness perception in neutral faces" has now been seen by 2 reviewers, and I include their comments at the end of this message. They find your work of interest but raised some important points. We are interested in the possibility of publishing your study in *Communications Psychology*, but would like to consider your responses to these concerns and assess a revised manuscript before we make a final decision on publication.

We therefore invite you to revise and resubmit your manuscript, along with a point-by-point response to the reviewers. Please highlight all changes in the manuscript text file.

Editorially, we consider it crucial that reviewer #2's methodological concerns, including the beta-band coherence analysis and the choice of fNMES time window, are thoroughly addressed. Reviewer #2 also made some statistical suggestions that align with our journal's standard, such as the inclusion of effect sizes. Please include effect size measures as well as Bayesian statistics for the null results (e.g., no difference between early and late fNMES).

Please ensure you follow our statistical guidelines when reporting statistics (<https://www.nature.com/commspsychol/submit/submission-guidelines#statistical-guidelines>). Please note in particular our requirements for the reporting and interpretation of null-results. Non-significant findings derived from null-hypotheses significance tests should be reported in full, but may not be interpreted. Where you interpret null results, this interpretation must be based on Bayes Factors or equivalence tests.

I am attaching an Editorial Requests Table that details critical reporting requirements for the revised manuscript. Please attend to each item and ensure your manuscript is fully compliant. If your revised manuscript is not aligned with these requests on major issues, such as those concerning statistics, it may be returned to you for further revisions without re-review.

Please submit the following items:

- Revised manuscript
- Point-by-point response to the referees' comments
- Cover letter (as a separate document)

- <https://www.nature.com/documents/nr-reporting-summary.zip>>Nature Research Reporting Summary
- <https://www.nature.com/documents/nr-editorial-policy-checklist.pdf>>Editorial Policy Checklist
- Completed Editorial Request Table (attached).

via this link: Link Redacted .

Additional guidance is available in our style and formatting guide Communications Psychology formatting guide.

Best regards,

Troy Lui

Troy Lui, PhD
Associate Editor
Communications Psychology

REVIEWER EXPERTISE:

Reviewer #1: face, ERP/EEG, Emotion

Reviewer #2: face stimulation

REVIEWER REPORTS:

Reviewer #1 (Remarks to the Author):

Review of "The face says it all: electrical stimulation of smiling muscles reduces visual processing load and enhances happiness perception in neutral face"

I very much like the idea, procedure and results of this paper. I am generally very positive about it. I appreciate how transparent they are with the methods, analysis and data through their Open Science Framework site. The study is methodologically sound, by employing simultaneously facial neuromuscular electrical stimulation (fNMES) and EEG recordings on an emotion recognition task, it provides solid and novel evidence towards the classical theory of embodied cognition. The paper has an extremely clear methods description, analysis and results are clear, both behavioural and EEG. I am willing to support its publication, I only have a few comments more relate to the form of telling the story than to the content. only

The introduction provides a complete revision of the literature and does a good job acknowledging research, theories and highlighting gaps in data that fNMES can easily fulfil. However, at parts it feels a bit too length for the current journal. For example, some methodological aspects (timing choice lines 70-85; how can be combined with EEG 87-97; introduction and generalities on N170 98-125) can be shorten, similar with general introduction and so on.

The abstract requires some re writing to increase likeability and fluidity. The beginning feels a bit clunky and towards the end one feels repetition. A more fluid narrative would make it easier to read.

Abstract line 26. May be useful to specify you are referring to the 'visual' face onset or 'visual' face processing.

Line 177. the idea of action units is unclear.

Figure 1 add units to -250: 250 ms. 500:1000ms. The figure would benefit from a clear representation of the observer face stimulation. As a picture or schema of observers faces and placement of electrodes.

Reviewer #2 (Remarks to the Author):

The study explores facial feedback's role in emotion recognition using facial neuromuscular electrical stimulation (fNMES). Researchers stimulated the zygomaticus major (ZM) muscle—responsible for smiling—at key intervals to examine its interaction with visual processing. EEG measurements assessed neural responses, focusing on N170 amplitude and beta-band coherence between somatomotor and occipital cortices.

Key Findings:

- fNMES increased happy categorization of neutral faces during early (-250 to +250 ms) and late (500 to 1000 ms) stimulation.

- Early fNMES decreased N170 amplitude, indicating a reduced visual processing load.

- Late fNMES modulated beta-band coherence, but this did not predict emotion categorization.

These findings enhance embodied cognition theories and offer insights into facial feedback's influence on visual processing. However, questions remain about the underlying neural mechanisms, the generalizability of results, and practical implications for emotion modulation therapies.

Strengths

Innovative fNMES for Facial Feedback

fNMES offers precise muscle control, unlike pen-in-mouth tasks. Controlled stimulation timing differentiates early and late facial feedback effects.

Integration of EEG for Neural Correlates

The study employs N170 and beta-band coherence as objective markers, reinforcing the link between somatosensory feedback and visual emotion processing.

Rigorous Experimental Design

Two pilot studies optimized stimulus selection and fNMES timing, with pre-registered hypotheses and open data-sharing enhancing reproducibility.

Relevance to Theories of Embodied Cognition

The study supports predictive coding models, where facial feedback modulates emotional perception.

Weaknesses & Areas for Improvement

The authors hypothesized that late fNMES (500-1000 ms) would have a stronger effect on emotion categorization than early fNMES (-250 to +250 ms), but this was not supported. The underlying rationale for this hypothesis is not clearly articulated, and the study does not adequately justify why late feedback was expected to have a greater impact beyond referencing the general literature on spontaneous facial mimicry (SFM). Since fNMES effects occurred in both conditions, it remains unclear whether late fNMES does not confer additional benefits or whether the specific late time window (500-1000 ms) was suboptimal. A stronger theoretical justification for why late feedback was expected to be more influential would improve coherence, and future studies should refine the late fNMES timing by testing alternative time windows (e.g., 400–800 ms, 600–1100 ms).

The study interprets the reduction in N170 amplitude as a reduction in visual processing load, suggesting that facial feedback offsets the need for deeper visual analysis. However, alternative explanations, such as predictive coding violations, attentional shifts, or general sensory suppression, are not sufficiently discussed. Predictive coding models suggest that N170 reduction may reflect expectation violations—if fNMES creates a prior expectation of a happy face, then encountering a different expression may generate a prediction error signal. The study does not cite EEG literature on predictive coding violations or sensory suppression effects on N170, which would provide a stronger context for interpretation. Expanding the discussion to include predictive coding frameworks and citing EEG studies that explore N170 changes in expectancy violations and somatosensory integration would clarify the interpretation. Additional EEG markers (e.g., P300, late positive potential) to determine whether N170 changes reflect top-down expectancy violations rather than reduced visual load may be helpful.

Late fNMES modulated beta-band coherence between somatomotor and occipital cortices, but this did not predict emotion categorization. The authors suggest that the lack of a significant relationship may be due to insufficient statistical power, but it remains unclear whether this represents a true null effect or a methodological limitation. The study does not sufficiently justify why beta-band was the primary frequency of interest over gamma-band (associated with neural binding) or alpha-band (linked to attentional control). Single-trial variability was not analyzed, meaning that potentially meaningful relationships between coherence and behavior may have been masked by averaging. A stronger theoretical rationale for focusing on beta-band coherence, explore single-trial analyses to determine whether coherence at specific time points predicts behavioral choices, and multivariate approaches may help detect relationships between coherence patterns and emotion categorization.

The 500–1000 ms window for late fNMES was based on spontaneous facial mimicry (SFM) timing for happy faces (Pilot Study 2). However, the task included both happy and sad faces, meaning that the late fNMES window might not have been optimized for sad expressions. If SFM for sad faces occurs later than for happy faces, then the 500–1000 ms window may have been too early to capture late feedback effects for sad expressions. This could explain why late fNMES effects were not stronger despite the hypothesis predicting that facial feedback during SFM would have the greatest impact (SFM timing for multiple emotions, test alternative late time windows, and use facial EMG recordings to directly measure when spontaneous mimicry occurs for different emotions may be needed to explain this).

The study used FaceGen avatars, which lack naturalistic microexpressions, asymmetries, and movement dynamics. This may have influenced how participants perceived emotions, making it unclear whether fNMES effects would generalize to real-world settings. Natural human faces contain subtle movement cues that aid emotion perception, while artificial faces do not. Participants may process artificial faces differently than real faces, affecting facial feedback integration. The findings may not generalize to real-world face perception or social interactions and real human faces, use videos of dynamic facial expressions to test whether fNMES effects persist with more naturalistic stimuli, and implement eye-tracking to determine whether gaze patterns differ between FaceGen avatars and real faces may be helpful.

While the study reports statistically significant effects, it does not explicitly discuss effect sizes. Given the modest sample size ($N = 51$), there is a risk of overinterpretation. Effect sizes are not provided in key analyses, making it difficult to assess the practical significance of findings. The study does not quantify the likelihood of Type I or Type II errors, despite noting power limitations in some analyses. Reporting Cohen's d , η^2 , or Bayes Factors for all major findings, adding effect size annotations in figures (e.g., Figure 3b for happiness choice, Figure 6 for beta coherence), and using Bayesian analyses to quantify the strength of evidence for key effects would improve statistical transparency.

Version 1:

Decision Letter:

Dear Dr Baker,

Your manuscript titled "The face says it all: electrical stimulation of smiling muscles reduces visual processing load and enhances happiness perception in neutral faces" has now been seen by our reviewers, whose comments appear below. In light of their advice I am delighted to say that we are happy, in principle, to publish a suitably revised version in Communications Psychology.

We therefore invite you to revise your paper one last time to address the remaining concerns of our reviewers and a list of editorial requests. At the same time we ask that you edit your manuscript to comply with our format requirements and to maximise the accessibility and therefore the impact of your work.

EDITORIAL REQUESTS:

SUBMISSION INFORMATION:

OPEN ACCESS:

* DATA AVAILABILITY:

Link Redacted

Best regards,

Troy Lui

Troy Lui, PhD
Associate Editor
Communications Psychology

REVIEWERS' COMMENTS:

Reviewer #1 (Remarks to the Author):

I appreciate the changes in the manuscript, which improve readability and clarity. The changes in the figures provide additional clarity. I do not have any additional comments on this paper. I feel it has well addressed the reviewers' comments, and I recommend it for publication.

Reviewer #2 (Remarks to the Author):

The authors have addressed the suggestions appropriately.

Response to reviewers: The face says it all: electrical stimulation of smiling muscles reduces visual processing load and enhances happiness perception in neutral faces (COMMSPSYCHOL-24-0734-T)

Dear Dr Troby Lui and reviewers,

Thank you for your time and efforts in getting reviews for our submission. We thank the reviewers for their detailed feedback and constructive comments. We detail below our response to each point that the reviewers have made. The reviewers' comments are in black, whilst our responses are in red. Additionally, we provide the new text we have added into the manuscript directly in this letter in blue. Newly added text is also highlighted in the manuscript.

In response to points made by reviewer #2 and the editor, we want to emphasise that we have mainly used mixed models in our analysis, as they offer significant advantages in controlling for Type I errors and addressing non-independence compared to other methods, and that we report model fits and effect sizes as customary in the field. Because variance is partitioned differently in generalized linear mixed models, there is no universally accepted method for determining standard effect sizes for specific model terms like main effects or interactions (e.g., Rights & Sterba, 2019; and see this link on stats.exchange). Instead, R^2 values represent the proportion of variance accounted for by the model and are therefore regarded as effect size measures. However, we are also reporting an approximate mapping of Cohen's d for our key effects, using the formula $d = \beta / (\pi / \sqrt{3})$, as suggested by Borenstein et al. (2009, Converting among effect sizes, chapter 7). Be aware that this can produce both positive and negative Cohen's d values, indicating both the magnitude and the direction of the effects. We already reported in the previous version of the manuscript partial eta squared for the ANOVAs. We have now added Bayes factors for non-significant key effects. These were computed using the BayesFactor package in R. In addition, we have now added to our methods information a power analysis we used to inform our sample size. This was already included in our pre-registration, however we now include this in the participants section of the methods.

Reviewer #1

We thank the reviewer for their positive comments and agree that the introduction and abstract could be strengthened. Please see below how we addressed this.

"Some methodological aspects (timing choice lines 70-85; how can be combined with EEG 87-97; introduction and generalities on N170 98-125) can be shorten, similar with general introduction and so on".

We have now significantly shortened the introduction. We have highlighted the new shortened text in the manuscript.

"The abstract requires some re writing to increase likeability and fluidity. The beginning feels a bit clanky and towards the end one feels repetition. A more fluid narrative would make it easier to read".

We have now completely re-written the abstract to increase likeability and fluidity. The abstract now reads: Theories of embodied cognition suggest that after an initial visual processing stage, emotional faces elicit spontaneous facial mimicry (SFM), and that the accompanying change in proprioceptive

facial feedback contributes to facial emotion recognition. However, this temporal sequence has not yet been properly tested, given the lack of methods allowing to manipulate or interfere with facial muscle activity at specific time points. The current study (N=52, 28 female) investigated this key question using EEG and facial neuromuscular electrical stimulation (fNMES) – a technique offering superior control over which facial muscles are activated and when. Participants categorised neutral, happy, and sad avatar faces as either happy or sad, and received fNMES (except in the control condition) to bilateral zygomaticus major muscles during early visual processing (-250 to +250 ms of face onset), or later visual processing, when SFM typically arises (500 to 1000 ms after face onset). Both early and late fNMES resulted in a happiness bias specific to neutral faces, which was mediated by a reduced N170 in the early window. In contrast, a modulation of the beta-band (13-22 Hz) coherence between somatomotor and occipital cortices was found in the late fNMES, although this did not predict categorisation choice. We propose that facial feedback biases emotion recognition at different visual processing stages by reducing visual processing load.

“Abstract line 26. May be useful to specify you are referring to the ‘visual’ face onset or ‘visual’ face processing”.

Please see above comment about re-writing the abstract.

“Line 177. the idea of action units is unclear.”

We have now added the following text at the very top of page 9 in order to expand on the definition of Action Units. This process involves manipulating Action Units (AUs), which correspond to specific muscle (group) movements that contribute to facial expressions.

“Figure 1 add units to -250: 250 ms. 500:1000ms. The figure would benefit from a clear representation of the observer face stimulation. As a picture or schema of observers faces and placement of electrodes.”

We have now added the unit ‘ms’ to the figure, and also added a schematic showing the electrode placement in a new panel C. We also added the following text to the figure caption: c) schematic of electrode placement over the participant’s bilateral zygomaticus major muscles.

Reviewer #2

We thank the reviewer for their detailed comments and suggestions. Please see below how we have addressed each point.

“The authors hypothesized that late fNMES (500-1000 ms) would have a stronger effect on emotion categorization than early fNMES (-250 to +250 ms), but this was not supported. The underlying rationale for this hypothesis is not clearly articulated, and the study does not adequately justify why late feedback was expected to have a greater impact beyond referencing the general literature on spontaneous facial mimicry (SFM). Since fNMES effects occurred in both conditions, it remains unclear whether late fNMES does not confer additional benefits or whether the specific late time window (500-1000 ms) was suboptimal. A stronger theoretical justification for why late feedback was expected to be more influential would improve coherence, and future studies should refine the late fNMES timing by testing alternative time windows (e.g., 400–800 ms, 600–1100 ms).”

Many thanks for these important questions. We have rephrased the hypothesis section at the end of the introduction, to clarify why we expected larger effects for the late fNMES period (p. 7): We also

expected (H2) this effect to be larger for the late fNMES period, which overlaps with the time period when spontaneous facial mimicry to emotional faces typically arises (as also confirmed in our EMG pilot experiment), and therefore should be the most natural timing for facial feedback changes to occur during emotion processing.

We agree that future studies should refine the best timing for late fNMES, and have discussed this on p. 26: The reason we did not observe stronger effects in the later condition could be due to the fact that stimulation was not delivered in the optimal period. Indeed, we based our late stimulation period on a pilot study which only considered SFM to happy faces. As such, our late period might have been slightly ill-informed (i.e. did not reflect the timing of SFM in the case of sad faces). Future studies should refine the late fNMES timing by testing alternative time windows (e.g., 400–800 ms, 600–1100 ms). Another possibility is that SFM manifests at a subthreshold level prior to being observable in facial EMG data. As such, although in a statistical sense SFM occurred only after 500 ms following face onset, it is possible that undetectable state-changes in facial muscles already occur prior to this, and thus our late stimulation period was indeed too late. Future studies could investigate this further by providing fNMES, for example, from 200 ms onwards.

“The study interprets the reduction in N170 amplitude as a reduction in visual processing load, suggesting that facial feedback offsets the need for deeper visual analysis. However, alternative explanations, such as predictive coding violations, attentional shifts, or general sensory suppression, are not sufficiently discussed. Predictive coding models suggest that N170 reduction may reflect expectation violations—if fNMES creates a prior expectation of a happy face, then encountering a different expression may generate a prediction error signal. The study does not cite EEG literature on predictive coding violations or sensory suppression effects on N170, which would provide a stronger context for interpretation. Expanding the discussion to include predictive coding frameworks and citing EEG studies that explore N170 changes in expectancy violations and somatosensory integration would clarify the interpretation. Additional EEG markers (e.g., P300, late positive potential) to determine whether N170 changes reflect top-down expectancy violations rather than reduced visual load may be helpful.”

Thank you for these suggestions. We agree that providing an alternative explanation for our findings (e.g. in the framework of predictive coding) enhances the manuscript. Please note that we already included a paragraph in the introduction (page 6) in which we introduce predictive coding, and cite previous work on expectancy violations and the N170. The text reads as follows: Modulations in N170 amplitude due to changes in facial feedback can also be contextualized within the framework of predictive coding (Friston & Kiebel, 2009). The brain continuously generates predictions about sensory inputs and compares these with actual incoming sensory data (Brodski-Guerniero et al., 2017). If there is a mismatch, it results in a prediction error signal, which indicates that something unexpected has occurred. The internal model is then adjusted, so as to perpetually minimise these prediction errors. Indeed, N170 amplitude has previously been demonstrated to be sensitive to unexpected perceptual events (Robinson et al., 2020), and the degree to which expectations are violated has a graded effect on N170 amplitude (the larger the violation, the larger the N170). Activations of ZM may therefore provide a certain context in which a happy face is predicted. If a different face, for example, is then presented (a violation of this prediction), then the resulting prediction error may manifest as an increase in N170 amplitude (Sel et al., 2015).

In addition, we also already include the following in the discussion (page 28): An alternative explanation for this modulation in N170 is provided by predictive coding, a theoretical framework aiming to explain how the brain processes sensory information (Friston & Kiebel, 2009). Stimulation of ZM may lead to the prediction that a happy face is to be presented, and a prediction error would

occur if the presented face violates this prediction. Indeed, during early fNMES, N170 was greater (more negative) to sad than happy faces (see figure S3 in Supplementary materials). As such, the unexpected occurrence of a sad face (given the activation of the participant's smiling muscles) resulted in a larger prediction error, reflected in a larger N170. Indeed, N170 amplitude has previously been demonstrated to be sensitive to unexpected perceptual events (Robinson et al., 2020), and the degree to which expectations are violated has a graded effect on N170 amplitude (the larger the violation, the larger the N170). When prediction errors are small or absent, the N170 amplitude will become relatively smaller, which is what we found in response to happy faces (and neutral faces labelled as happy) during fNMES to the ZM muscles (the perceived face was consistent with the provided facial feedback context).

Regarding the comment about examining additional EEG markers; the study implemented a data driven approach, in that we contrasted several conditions using a mass-univariate analysis. The mass-univariate approach reduces the risk, common in ERP studies, to find significant-but-bogus effects (see Luck & Gaspelin, 2018, Psychophysiology: How to Get Statistically Significant Effects in Any ERP Experiment (and Why You Shouldn't)). Our mass univariate analyses identified only significant differences in the range of the N170 at temporal-occipital sites (see figure 4 on page 21). As such, these analyses did not identify any significant interactions in the P300/LPP range. If differences were to be observed, we would have investigated further.

"Late fNMES modulated beta-band coherence between somatomotor and occipital cortices, but this did not predict emotion categorization. The authors suggest that the lack of a significant relationship may be due to insufficient statistical power, but it remains unclear whether this represents a true null effect or a methodological limitation. The study does not sufficiently justify why beta-band was the primary frequency of interest over gamma-band (associated with neural binding) or alpha-band (linked to attentional control). Single-trial variability was not analyzed, meaning that potentially meaningful relationships between coherence and behavior may have been masked by averaging. A stronger theoretical rationale for focusing on beta-band coherence, explore single-trial analyses to determine whether coherence at specific time points predicts behavioral choices, and multivariate approaches may help detect relationships between coherence patterns and emotion categorization".

Thank you for these comments. To better explain why we focused on coherence in the beta range, we have now added the following text to the introduction on page 5 and to the discussion on page 28: Coherence in the beta-band has been suggested to represent long-distance EEG connectivity for the processing of stimuli with affective value (Aftanas et al., 2002).

We have adopted a rather data-driven approach (similar to the ERP analyses) to identify the frequency range of interest for the coherence analyses. We implemented Monte Carlo based cluster permutation tests on spectra between 5 and 40 Hz. As such, our analyses did consider the alpha and gamma range. Given that we identified significant differences in coherence only in the beta band, we restricted our introductory text to reflect this. However, we have now added the following text to the discussion on page 29: Although our analyses identified fNMES effects specifically in the beta-band, future studies might wish to consider exploring modulations of coherence in other frequencies during facial feedback manipulations. For example, gamma band activity has been found to represent the binding of stimulus features with top-down information (Widmann et al., 2007).

Finally in regards to trial-by-trial variability in coherence, we would like to clarify that single-trial coherence estimates cannot be computed. Coherence is measured across trials, and for single trial coherence, the value is 1. Please see the following link on the Fieldtrip mailing list, where Jan Mathijs

(one of the core developers of Fieldtrip) describes that single-trial coherence estimates are not computable. <https://mailman.science.ru.nl/pipermail/fieldtrip/2012-November/005918.html>

“The 500–1000 ms window for late fNMES was based on spontaneous facial mimicry (SFM) timing for happy faces (Pilot Study 2). However, the task included both happy and sad faces, meaning that the late fNMES window might not have been optimized for sad expressions. If SFM for sad faces occurs later than for happy faces, then the 500–1000 ms window may have been too early to capture late feedback effects for sad expressions. This could explain why late fNMES effects were not stronger despite the hypothesis predicting that facial feedback during SFM would have the greatest impact (SFM timing for multiple emotions, test alternative late time windows, and use facial EMG recordings to directly measure when spontaneous mimicry occurs for different emotions may be needed to explain this).”

We agree that our pilot study using facial EMG in response to happy and neutral faces only may have resulted in a suboptimal late fNMES time window. We already include in the discussion on page 26 the following text: The reason we did not observe stronger effects in the later condition could be due to the fact that stimulation was not delivered in the optimal period. Indeed, we based our late stimulation period on a pilot study which only considered SFM to happy faces. As such, our late period might have been slightly ill-informed (i.e. did not reflect the timing of SFM in the case of sad faces). Future studies should refine the late fNMES timing by testing alternative time windows (e.g., 400–800 ms, 600–1100 ms).

“The study used FaceGen avatars, which lack naturalistic microexpressions, asymmetries, and movement dynamics. This may have influenced how participants perceived emotions, making it unclear whether fNMES effects would generalize to real-world settings. Natural human faces contain subtle movement cues that aid emotion perception, while artificial faces do not. Participants may process artificial faces differently than real faces, affecting facial feedback integration. The findings may not generalize to real-world face perception or social interactions and real human faces, use videos of dynamic facial expressions to test whether fNMES effects persist with more naturalistic stimuli, and implement eye-tracking to determine whether gaze patterns differ between FaceGen avatars and real faces may be helpful”.

Whilst we agree that using dynamic stimuli may afford a more ecologically valid approach and a richer visual signal, we opted for static images of faces so as to achieve sufficient temporal control for the averaging of face-specific evoked responses. In addition, by using avatar faces, we were able to tightly control low-level visual features, features of which could contribute unwanted variability in the EEG response (see <https://pubmed.ncbi.nlm.nih.gov/17629569/> and <https://www.nature.com/articles/s41598-018-20467-1> . Although we acknowledge that the responses to avatar faces may indeed differ from real life faces (see <https://pmc.ncbi.nlm.nih.gov/articles/PMC7235958/>), we would argue that the responses are similar enough, and are sufficient in eliciting face-specific EEG activity. Indeed, the avatar expressions were created based on the facial action coding system (FACS, the golden standard for defining facial movement), and it is very common in the literature to use avatar faces in place of real faces to elicit embodied face processing responses that can be measured in the face (e.g. with EMG) and the brain (e.g. with ERPs). We agree that one should consider the potential limitations of using avatars, and so we now include the following text to the discussion on page 29: It is also important to consider that avatar faces, although allowing for tight control of low-level visual features, might differ from real pictures of facial expressions in their eliciting of processes involved in emotional recognition. As such, future studies might wish to present real-life, even dynamic facial expressions so as to maximise the generalizability of our findings.

“While the study reports statistically significant effects, it does not explicitly discuss effect sizes. Given the modest sample size ($N = 51$), there is a risk of overinterpretation. Effect sizes are not provided in key analyses, making it difficult to assess the practical significance of findings. The study does not quantify the likelihood of Type I or Type II errors, despite noting power limitations in some analyses. Reporting Cohen’s d , η^2 , or Bayes Factors for all major findings, adding effect size annotations in figures (e.g., Figure 3b for happiness choice, Figure 6 for beta coherence), and using Bayesian analyses to quantify the strength of evidence for key effects would improve statistical transparency.”

Thank you for making these suggestions. We have provided a response to this at the top of this document. We also include the same text below.

We want to emphasise that we have mainly used mixed models in our analysis, as they offer significant advantages in controlling for Type I errors and addressing non-independence compared to other methods, and that we report model fits and effect sizes as customary in the field. Because variance is partitioned differently in generalized linear mixed models, there is no universally accepted method for determining standard effect sizes for specific model terms like main effects or interactions (e.g., Rights & Sterba, 2019; and see this link on stats.exchange). Instead, R^2 values represent the proportion of variance accounted for by the model and are therefore regarded as effect size measures. However, we are also reporting an approximate mapping of Cohen’s d for our key effects, using the formula $d = \beta / (\pi / \sqrt{3})$, as suggested by Borenstein et al. (2009, *Converting among effect sizes*, chapter 7). Be aware that this can produce both positive and negative Cohen’s d values, indicating both the magnitude and the direction of the effects. We already reported in the previous version of the manuscript partial eta squared for the ANOVAs. We have now added Bayes factors for non-significant key effects. These were computed using the BayesFactor package in R.